# Human Milk Oligosaccharides (HMOs) and Infant Microbiota: A Scoping Review

**DOI:** 10.3390/foods10061429

**Published:** 2021-06-20

**Authors:** Cristina Sánchez, Cristina Fente, Patricia Regal, Alexandre Lamas, María Paz Lorenzo

**Affiliations:** 1Center for Metabolomics and Bioanalysis (CEMBIO), Facultad de Farmacia, Campus Montepríncipe, Universidad San Pablo-CEU, Boadilla del Monte, 28668 Madrid, Spain; c.sanchez127@usp.ceu.es (C.S.); pazloga@ceu.es (M.P.L.); 2Department of Analytical Chemistry, Nutrition and Bromatology, Santiago de Compostela University, 27002 Lugo, Spain; patricia.regal@usc.es (P.R.); alexandre.lamas@usc.es (A.L.)

**Keywords:** breast milk, breastfeeding, human milk oligosaccharide, infant gut microbiota, microbiome

## Abstract

Human milk oligosaccharides (HMOs) are the third most abundant solid component of breast milk. However, the newborn cannot assimilate them as nutrients. They are recognized prebiotic agents (the first in the newborn diet) that stimulate the growth of beneficial microorganisms, mainly the genus *Bifidobacterium,* dominant in the gut of breastfed infants. The structures of the oligosaccharides vary mainly according to maternal genetics, but also other maternal factors such as parity and mode of delivery, age, diet, and nutritional status or even geographic location and seasonality cause different breast milk oligosaccharides profiles. Differences in the profiles of HMO have been linked to breast milk microbiota and gut microbial colonization of babies. Here, we provide a review of the scope of reports on associations between HMOs and the infant gut microbiota to assess the impact of HMO composition.

## 1. Introduction

The advantages of breastfeeding are unanimously defended by all health establishments [1]. The more the composition of human milk is known, the better the benefits for the child and the mother are understood. Non-nutritive carbohydrates in human milk (called human milk oligosaccharides, HMOs) are the third solid component in percentage after lactose and lipids and practically equal to proteins. The breastfed baby can consume up 10 g per day of non-nutritive carbohydrates [2]. The very energy-expensive synthesis of HMOs can only be understood, in evolutionary terms, due to its important contribution to the child’s survival in the first months of life. The mother confers protection to the newborn through breastfeeding, and HMOs are part of these breast milk compounds involved in that protection.

The HMOs (up to 200) can be composed of five different monosaccharides: glucose (Glc), galactose (Gal), N-acetyglucosamine (GlcNAc), fucose (Fuc), and the sialic acid (Sia) derivative N-acetyl-neuraminic acid (Neu5Ac) [3]. They all have a lactose or an N-acetyllactosamine [4] at their reducing end. Lactose can be fucosylated or sialylated into small oligosaccharides or elongated with disaccharides to form larger oligosaccharides ranging in size from 3 to 32 sugars [3]. The maternal genetic variants influence the structure of oligosaccharides encoded by genes associated with the expression of the Lewis blood group system; secretor (Se) and Lewis (Le) blood group genes are implicated. The Se gene that encodes the α 1–2-fucosyltransferase enzyme (FUT2) in non-reducing end Gal and mother is called the secretor. The Le gene encodes α 1–3/4 fucosyltransferase (FUT3) in the Glc end and results in the presence of Le α -sugars in secretor (Se+) or non-secretor (Se-) milk. Based on the expression of FUT2 and FUT3, mothers Se+ and Le+ can secrete all HMOs; Se+ and Le- can secrete 2′-fucosyllactose (2′FL), 3′-fucosyllactose (3′FL), lacto-N-fucopentaose I (LNFP I), and lacto-N-fucopentaose III (LNFP III); Se- and Le+ can secrete 3′FL, lacto-N-fucopentaose II (LNFP II), and LNFP III; and Se- and Se- can secrete 3′FL, LNFPIII, and LNFP-V [5]. Consequently, the HMOs breast milk composition varies significantly between these four groups [6]. Regarding sialylation, there may also be slight differences due to variations in sialyltransferases’ expression [3]. The Gal end can be sialylated in α2-3 or α2-6 or elongated in β1-3 by lacto-Nbiose I (Galβ1,3-GlcNAc) or in β1-6 by N-acetyl-lactosamine (Galβ1, 4GlcNAc). Other ramifications may occur with Fuc, Sia derivative Neu5Ac, and/or N-acetyllactosamine. Thus, HMOs are named fucosylated neutral, non-fucosylated neutral, and sialylated HMOs [3]. In addition to genetics, other maternal factors such as parity and mode of delivery, age, diet, and nutritional status or even geographic location and seasonality can influence HMOs’ breast milk profiles [7].

HMO profiles influence the microbiota of breast milk and the infant gut. Once ingested, breast milk oligosaccharides reach the distal area of the small intestine and colon practically intact. They are recognized prebiotic agents (the first in the newborn diet) that stimulate the growth of beneficial microorganisms, mainly *Bifidobacterium* genus (dominant species in breastfed infants gut) and, to a lesser extent, some strains of *Bacteroides* and *Lactobacillus*. As these bacteria specifically express sialidases and fucosidases, oligosaccharides promote the growth of these strains over other bacteria that cannot use HMOs as an energy source [8]. On the other hand, maternal oligosaccharides increase the adhesion of the selected strains to the intestinal mucosa, improving their persistence in the mucosa and increasing the anti-inflammatory effects on the human intestine [9]. In addition, HMOs can protect infants by reducing the incidence of intestinal diseases, acting as antiadhesives in interactions with the host, in two ways: selectively binding to pathogens or their toxins, then inhibiting their adherence to glucan ligands on the mucosal cell surface [10], or they can bind themselves to glucocalyx on the surface of epithelial cells [3]. The partial metabolization of oligosaccharides gives rise to “postbiotic” compounds that stimulate the growth of other types of butyrate and propionate-producing microbiota. These short-chain fatty acids have a trophic effect on the intestinal barrier, stimulating mucin release and modulating the immune system, promoting immune tolerance [11]. The HMOs-consuming bacteria also inhibit the pathogenic bacterium colonizing the intestine by reducing nutrient availability and the production of antimicrobial substances. A direct bacteriostatic action of oligosaccharides in breast milk has been demonstrated in the case of group B Streptococcus, which cannot proliferate in a medium with specific non-sialylated HMOs [12].

The primary objective of this scoping review is to provide a critical appraisal of known associations between HMO composition and the infant gut microbiome to assess the impact of HMO composition on the child microbiome. There are limited reports on the associations between HMOs and child gut microbiota in the literature. A brief discussion on hypotheses around possible mechanisms by which the HMO profile may influence the microbiome will be included.

## 2. Methodologies

### 2.1. Search Strategy and Inclusion Criteria

A scoping review was used to synthesize the evidence and assess the scope of the literature on the topic. This review was based on the PRISMA Extension for Scoping Reviews (PRISMA-ScR) approach [13]. Identifying a research question; identifying relevant studies; selecting relevant studies; graphing the data; and collecting, summarizing, and reporting the results [14] are described in more detail below.

The research question was “What is known about the influence of oligosaccharide profile in human milk on the infant gut microbiota?”.

A comprehensive literature search of PubMed, Scopus, and Web of Science Core Collection was performed in April 2021 and was limited to articles published in English from ten years ago. Text words and controlled vocabulary for four concepts, breastmilk, oligosaccharides, infant gut, and microbiome, within the titles, abstracts, and keywords of articles were used.

Original research papers were revised that investigated associations between HMOs, as free glycans, (profiles or maternal secretor status, MSeS, phenotypically analyzed) and the gut microbiome of breastfed infants. Outcomes that were assessed included any alterations of the composition and function in the infant gut microbiota, from neonates or infant until one year aged, and their relationship with absolute concentrations of any HMO or total HMO content. Studies published as full-length articles, excluding conference abstracts, books, editorials, and letters to the editor, were selected. Randomized controlled trials, prospective, cohort studies, and cross-sectional observational studies examining breastfeeding infants were screened. Reviews were excluded.

### 2.2. Article Screening and Data Abstraction

Titles and abstracts of all papers were assessed for their potential relevance according to the inclusion and exclusion criteria. Data were extracted from the full-text papers and subsequently reviewed. Studies were initially appraised individually before comparing and summarizing the findings for links between the HMO profile and the infant gut.

## 3. Results

### 3.1. Synthesis

A total of 308 records were identified in the databases applying the exclusion and inclusion criteria and after duplicate removal. Once the title and abstract had been studied, we chose 50 papers that were submitted to the full-text evaluation. In total, 17 publications were included in this scoping review. The final search results were shown in the PRISMA flow chart (Figure 1).

Data abstraction details including a summary of the study design, population and sample number, oligosaccharides in breast milk and fecal microbial analysis methodology, outcome observed, and the main finding for all studies included in the review are presented in Table 1.

The studies included in the present review were conducted in Brazil [15], United States [28,29,30], United Kingdom [16], Netherlands [17,20], Italy [18,31], Belgium [18], Canada [19], Kenya [21], China [22], Finland [23], Gambia [24], Australia [25,27], and Japan [26].

Two of the 17 studies were cross-sectional [15,21]. Paganini’s work [21] assessed the effects of secretor status on the maternal and infant gut microbiota in a cross-sectional analysis at baseline of one intervention trial. The work of Davis (2016) is a sub-study embedded within a randomized trial to investigate the effects of pre-natal and infancy nutritional supplementation on infant immune development [24].

Most of the studies were prospective observational cohort studies [16,17,19,20,22,23,25,27,28,31], and another was a proof-of-concept study [29]. We have also included one classified as a quasi-experimental cohort study [30]. Actually, two HMOs (2′FL and LNnT) are available to use in intervention trials to know their impact on the establishment of the gut microbiota. In this review, we included one randomized controlled clinical trial [18] with oligosaccharides from breast milk and another with a galacto-oligosaccharides (GOS) [26]. Trials with bovine oligosaccharides have been excluded. In the De Leoz (2015) study [29], the HMOs analysis was carried out on fecal samples. MSeS was determined by the presence/absence of different fucosylated HMO [15,16,22], by determination of relative HMOs abundance [19,25,28], or by HMO quantification [23]. A hemagglutination inhibition technique was used in one work [27], and genotyping was used in the Lewis study [24].

### 3.2. Analysis of Methodologies

HMO analysis was performed by capillary electrophoresis with laser-induced fluorescence (CE-LIF) in only one study [19]. In the rest of the works, the analysis of the profile of HMOs was carried out with various chromatographic and detection techniques: high performance (HP) anion exchange chromatography–pulsed amperometry detection (HPAEC-PAD) [21,23,31], porous graphitized carbon-ultra HPLC-mass spectrometry (PGC-UPLC-MS) [17], UPLC [20], HPLC-MS [15,16,19], matrix-assisted laser desorption/ionization–time of fight-mass spectrometry (MALDI-TOF-MS) [23], LC-QTOF-MS [22,24], or Nano-HPLC-chip/TOF-MS [25,28,29,30]. Detailed analysis of oligosaccharides remains very challenging due to the variety of hydrophilicity and ionization properties of the different structures [29].

Research papers that investigated associations between maternal status (Se+ or Se-) and infant microbiome were the majority [15,16,19,21,22,23,24,25,27,28]. One of them examines the intestinal flora at two years of age [27].

Apart from MSeS, eight studies quantified individual HMOs [16,17,19,20,21,29,30,31]. Quin et al. [19] also analyzed HMOs containing sulfate and/or phosphate groups. The following HMOs were often quantified: difucosyllacto-N-hexaose (DFLNH), difucosyllactose (DiFL), difucosyllacto-N-hexaose (DFLNH), difucosyllacto-N-tetraose (DFLNT), disialyllacto-N-hexaose (DSLNH), disialyllacto-N-tetraose (DSLNT), 2′FL, 3′FL, fucodisialyllacto-N-hexaose (FDSLNH), fucosyllacto-N-hexaose (FLNH), fucosyllacto-N-hexaose III (FLNH III), lactodifucotetraose (LDFT), lacto-N-fucopentaose (LNFP), LNFP I, LNFP II, LNFP III, lacto-N-hexaose (LNH), lacto-N-teraose (LNT), lacto-N-neohexaose (LNnH), lacto-N-neotetraose (LNnT), monofucosyllacto-N-hexaose III (MFLNH III), 3′-sialyllactose (3′SL), 6′-sialyllactose (6′SL), sialyllacto-N-tetraose (SLNT), sialyllacto-N-tetraose a (SLNTa), sialyllacto-N-tetraose b (LSTb), sialyllacto-N-tetraose c (SLNT-c), and sialyllacto-N-tetraose d (LSTd).

Six works were longitudinal studies investigating the establishment of infant gut microbiota in relation to changes in breastmilk HMO composition [17,22,24,25,28,29]. One study analyzed the HMOs only in fecal samples [29].

Sequencing directed at different validated hypervariable regions of the 16S rRNA gene was used in most of the studies: not reported [25], V1–V2 [26], V3 [18], V4 [15,17,20,22,24,28], V3–V4 ([19,21,23], or V6–V8 [27]. In Masi’s study [16], metagenomic sequencing was performed, and in other research work, the quantitative polymerase chain reaction (PCR) served as a technique for the analysis of bifidobacterial species [31]. The sequencing of 16S rRNA gene amplicons has been a very popular approach to assess microbial communities in feces and other human matrices in the last decades [32,33]. Human feces are high microbial load samples, and both sample processing and primer selection largely impact 16S gene-based profiling results [34,35,36,37]. In this context, the protocol to extract DNA should be selected upon not only biomass abundance but also the expected gut microbiota composition. For instance, DNA extraction with no bead-beating step resulted in the absence of bifidobacteria in the sequence data, even when using optimized primers [34]. In Table 1, an overview is shown of different hypervariable regions of the 16S rRNA gene that have been targeted in gut microbiota studies, using different primer pairs. The choice of the 16S rRNA region can significantly affect the estimates of taxonomic diversity [38,39]. For instance, V2–V3 or V3–V4 regions compute a similar number of reads per phyla, but at lower taxonomic ranks, the differences become larger [38]. Interestingly, the literature indicates that bifidobacteria are often neglected by several common primer pairs [36]. For example, common primers targeting the V1 region have usually poor coverage of *Bifidobacterium*, while those targeting V4 will likely cover *Bifidobacterium* but not *Cutibacterium* [33]. To overcome these discrepancies and avoid biases, a careful selection of DNA extraction protocol and primer pairs is highly recommended.

## 4. Discussion

The infant gut microbiota is established in the first thousand days of life [40]. This colonization process is influenced even by prenatal factors. However, some scientists argue that evidence in support of the in utero colonization hypothesis is extremely weak [41]. Therefore, postnatal factors seem to be the most important factors influencing bacterial gut colonization. Mode of delivery is generally accepted as a significant factor associated with initial gut colonization. However, results are still inconclusive, and some research suggests that infant microbiota undergoes substantial reorganization during the first months of life [42]. The colonization of the newborn intestine begins with the microbiota of the birth canal in vaginal delivery, or nosocomial microbiota in cesarean section, as well as the bacteria transferred from mother’s milk. Enterobacteriaceae, *Streptococcus*, *Enterococcus*, and *Staphylococcus* are among the first colonizers. The growth of these facultative anaerobes creates a reduced oxygen environment that allows the expansion of obligate anaerobes such as *Bifidobacterium*, *Bacteroides,* or *Clostridium* in the next days [43]. After birth, the most influential factor of intestinal colonization is feeding practices. As an example, formula-fed infants had an overrepresentation of *Clostridium difficile* in comparison to breastfed infants [44]. HMOs resist gastric acidity, hydrolysis by host enzymes, and gastrointestinal absorption and due to their probiotic activity can be used by infant gut beneficial microorganisms as an energy source [45]. Among the bioactive compounds of human milk, HMOs have been reported to have the greatest influence on the infant’s gut microbiota shaping [42].

The HMOs are synthesized in the mammary gland. Depending on the expression of active fucosyl and sialyltransferases, more than 200 structurally distinct oligosaccharides can be generated. Based on the expression of FUT2 and FUT3, mothers can be divided into four groups: Se+Le+, Se+Le-, Se-Le+, and Se-Le- [5]. Genetic variants of the mothers are one of the main factors influencing HMOs profile, but other conditioning factors such as the mothers’ age, way of life, or even seasonal factors [24] or geographical origin may also be important [46]. However, more detailed assessment of nutrient intake during lactation may be required to identify (or exclude) dietary effects on HMO composition [47]. These differences in the HMOs composition have been related to the microbiome of breast milk and also to the microbiota that colonizes the gut of the breastfed child.

In this review, we have compiled the information available about the relationship between maternal status, the profile of HMOs, or the presence or absence of certain HMOs and the establishment of the microbiota in the baby’s intestine. It is not clear yet how interlaboratory differences in the analytical techniques can influence the results reported [47], but most studies included in this review focused on how the MSeS affects the infant gut microbiota [15,16,19,21,22,23,24,25,27,28]. The main interpersonal variation in the composition of HMOs is based on the secretor status of the woman. Se+ mothers have higher total HMOs concentrations than Se- mothers (median: approximately 10 vs. 5 g/L total HMO) [48]. The absence of 2′FL and other fucosyl-HMOs explains the lower total amount of HMO in the milk of women Se- [5]. However, all individual HMOs also differed by secretor status, except for disialyllacto-N-tetraose (DSLNT) [7].

### 4.1. What Impact Does MSeS Have on the Infant Gut Microbiota?

#### 4.1.1. Considering the Mode of Birth

Fucosylated oligosaccharides α1-2 are degraded by glycosyl hydrolase enzymes encoded by *Bifidobacterium* (*B. longum*, *B.bifidum*, *B. breve*) and *Bacteroides* (particularly *B. fragilis*) strains commonly present in the babies gut [49]. Children born by cesarean section generally have a lower abundance of these bacterial species [42]. In the Korpela cohort [23], newborns vaginally delivered, from Se+ and Se- mothers, did not present differences in their microbiota. All cesarean-born infants had significant reductions in the relative abundance of *Bacteroidetes* and an increase of Firmicutes. In the infants of Se+ mothers, a more modest deviation in microbiota composition is detected. Babies born by cesarean section had a lower presence of *Actinobacteria*, mainly *Bifidobacterium*, and those from Se- mothers had more *Enterococcus lactis*. According to the authors, this microorganism would opportunistically fulfill the niche available in the baby’s gut with fewer bifidobacteria. Lewis et al. [28] found similar results with higher abundance of streptococci, also belonging to the Lactobacillales order, in babies of Se- mothers. Furthermore, cesarean-born infants of Se+ mothers had significantly increased relative abundance of *Verrucomicrobia* (*Akkermansia muciniphila*) [23], which can degrade HMOs [50]. This can strengthen the gut barrier and likely contributes positively to infant gut health [51]. These facts can indicate that MSeS may be an important factor mainly among infants with otherwise compromised microbiota. Although Tonon’s [15] cross-sectional study did not compare the infant gut microbiota based on MSeS, it was included because corroborates the results obtained by the Korpela study. They did not observe differences in the relative abundance of bifidobacteria (except for *B. longum*) but reported significantly higher abundances of *Akkermansia muciniphila* in the gut microbiota of children born by cesarean section. The significantly higher abundance of *Akkermansia*, observed only in cesarean-section-born infants, was a finding of both studies. Even considering ethnicity and environmental factors (Nordic and Brazilian population), which strongly associated with microbial composition [52], maternal Se+ status equates the intestinal flora of a child born vaginally or by cesarean section.

#### 4.1.2. Regardless of the Mode of Delivery

Only two studies [21,25] found no significant differences in gut microbiota comparing children breastfed by Se+ or Se- mother. In the Paganini cross-sectional study, most of the women were Se+ and showed significant differences in concentration of HMO among mothers Se+ or Se-. However, there were no significant differences in the general composition of the intestinal microbiota, phylogenetic diversity, abundance of taxa of primary interest, or abundance of enteropathogens, with the exception of a greater abundance of *C. perfringens* among Se- mothers. The study reports that total HMO concentrations decrease during the course of lactation, which agrees with other research [53]. In Underwood’s study [25], 29 preterm BF infants were supplemented with *B. breve*. Stool sampling was performed near the time of probiotic initiation and again three weeks later. An increase in Enterobacteriaceae over time was pronounced in this cohort. Children were divided into ‘‘responders’’ and ‘‘nonresponders’’ when having few bifidobacteria in the second stool sample. Nonresponders had significantly higher percentages of Enterobacteriaceae and Clostridiaceae than responders. Infants with secretor mothers, delivery type, and antibiotic treatment did not differ between responder and nonresponder infants. Higher percentages of total fucosylated HMOs and lower percentages of undecorated HMOs (those lacking both fucose and sialic acid) were found in the milk-fed nonresponder babies. MSeS was not a significant predictor of response to the administered probiotic. Although B. breve M16-V is a selective consumer of human milk oligosaccharides (most strains consume 3′FL and LNT but not 2′FL), the undecorated HMOs, aggressively consumed by all B. breve strains, determine the differences.

More studies found clear differences in the intestinal microbiota of children fed by Se+ and Se- mothers. In Lewis’s cohort [28], Se+ fed infants generally had higher relative amounts of *Bifidobacterium* and *Bacteroides* and lower levels of *Enterobacteriaceae*, *Clostridia,* and *Streptococci*. The secretor status and the levels of infant bifidobacteria are statistically dependent variables. The group of infants with high bifidobacteria levels received milk significantly higher in non-fucosylated neutral and α (1-2)-fucosylated HMOs. This was a longitudinal study in which the samples were taken at days 6, 21, 71, and/or 120 postpartum. Therefore, a delayed colonization by bifidobacteria was found in the gut microbiota of infants fed by Se-. The authors attribute this delay to the infant’s difficulties acquiring a species of bifidobacteria capable of consuming the specific oligosaccharides of the milk supplied by the mother. For the first time, the possibility of increasing bifidobacteria in risk populations, such as premature babies, is conceived by adding specific glycans. In Smith-Brown’s research [27], children’s fecal samples were taken at two or three years of age. The MSeS was not determined by analyzing HMOs, was determined by using the hemagglutination inhibition technique. Despite this, the MSeS showed its influence on the composition and function of the microbiota. The MSeS explained a large amount of variation in children’s fecal microbiota profiles only when the analysis was limited to children who had been exclusively breastfed in their first 4 months of life. This observation suggests that breastfeeding may be an important factor. Davis’s observational and longitudinal study of a cohort of 33 Gambian mother/baby pairs [24] collected samples at weeks 4, 16, and 20 postpartum. Total HMO decreased significantly from 4 to 20 weeks after delivery. Regardless of time, Se+ mothers had higher relative concentrations of total fucosylated HMO, but lower relative concentrations of undecorated and sialylated HMOs. However, the mothers’ HMO profiles were significantly affected by the seasonal changes in Gambia. Increased energy intake could explain the mothers’ ability to produce milk with greater HMO concentrations during lactation in the dry season. This study focused on how HMOs consumed by infants and the associated microbiota affect infant health outcomes. The health results are not the purpose of this review; however, some of these findings help us to understand the relationship between HMOs and the infant microbiota. *B. infantis* encodes several enzymes to metabolize the main glycans of breast milk, which suggests that the dominance of this bacteria in the intestinal community is driven by the availability of these substrates [54,55,56]. In fact, in Davis’s study [24], they found positive or negative correlations between the microbial strains and the different HMOs. Thus, *B. infantis* was the only (sub-) species positively correlated with LNnT, confirming previous reports that particular HMO species enhance *B. infantis* growth [57]. *B. longum* subsp. *longum* that lacks fucosylated HMO-related genes was negatively correlated with total fucosylation and positively correlated with LNnT abundance, confirming previous observations [28]. Hence, the microbiome’s ability to metabolize certain types of oligosaccharides depends on the presence of specific strains in infant gut microbiota. This strain variability may contribute to their inability to find functional differences in microbiomes between babies fed by mothers of different secretor status.

Bai et al. [22] also conducted a longitudinal study to investigate how the secretor status and glycans of breast milk affected the gut microbiota of infants in Chinese babies born vaginally, exclusively breastfed, that were not receiving antibiotics/probiotics or complementary foods. Therefore, interference with the intestinal microbiota could be ruled out. They determined by 16S rDNA amplicon sequencing the fecal microbiota of lactating babies at four time points (days 6, 42, 120, and 180 after birth). Changes in the abundance of HMO and fucosylated N-glycans in milk at different stages, during 6 months of lactation, according to the condition of maternal secretor, were detected. The richest HMO content was observed in the colostrum, followed by a strong decrease after 42 days of lactation, also for most fucosylated HMOs, regardless of Se+ or Se- status. This was in agreement with the study by Thurl et al. [58], but Bai et al. [22] used MS to quantify all HMOs with 1,2-fucosylgalactose epitopes, providing a clearer picture of the relationship between HMOs and fucosyltransferase. The authors studied the colostrum microbiota, where no differences were observed between different MSeS. For this reason, the effects of the milk glycobiome on the intestinal microbiota stand out even more, taking the secretor type as a main influencing factor. Babies fed with Se- breast milk exhibited a highly fluctuating pattern throughout the six months of lactation. Thus, similarly to the study by Lewis et al. [28], *bifidobacterial* established earlier and in higher amounts in Se+-fed infants, and the relative abundances of this genus continued to increase beyond 180 days of lactation in the Se+ group. In the babies of Se- mothers, the abundance of bifidobacterial was lower, but it also increased during lactation. Regarding the relative abundance of species such as *B. breve*, capable of degrading α1-2 fucosylated [49], it was more abundant in infants fed Se+. Species in Masi’s work [16], regardless of secretor status, showed a positive correlation with sialylated HMO. Curiously, Bai et al. [22] observed the highest abundance of *Staphylococcus epidermidis* in the intestine of infants fed by Se+, the lowest presence of *Bacteroidetes* spp., also glycan consumers, in Se- babies, and a pattern of increasing levels of *Lactobacillus* spp. in babies of both groups, in the last stage of lactation (days 120 and 180). Although the order of *Lactobacillales* was significantly more abundant in the Se+ group, it was correlated with increasing levels of milk fucosylation glycoproteins. These results suggest a time-dependent expansion of the intestinal microbiota, finding no significant differences at 180 days in the values of the Shannon index (that reflects alpha diversity) between Se+ and Se-.

In the research of Quin et al. [19], although maternal genetics had modest effects on the infant fecal microbiome, the status was associated with Enterobacteriaceae in infants, suggesting a defining role of genetics in the establishment of early colonizers in infants. In this study, in addition to genetic factors, maternal dietary intake during lactation appears to influence the community composition of the infant microbiome. Fruit intake and unsaturated fatty acids in breast milk were positively correlated with an increased absolute abundance of numerous HMOs, including the 16 sulfonated HMOs identified here in humans for the first time. CE-LIF analysis has the possibility to detect 13 highly charged HMOs containing phosphate or sulfate. Red meat contains considerable amounts of Neu5Gc, associated with high abundance of *Bacteroides* genus [59]. Neu5Gc was incorporated into HMOs and correlates with *Bacteroides* spp. abundance in infant stool. Alongside Neu5Gc, the study shows that fucose levels in breast milk are associated with *Bacteroides* and *Escherichia* spp. in infant stools. Finally, and unexpectedly given that most *Lactobacillus* spp. do not grow well on HMOs, research found that *Lactobacillus* spp. in infant stools is correlated with total galactose concentration in sulfonated milk oligosaccharides [19].

In 2011, the pilot study of Coppa [31] analyzed oligosaccharides qualitatively and quantitatively, depending on the expression of Se and Le gene, and determined their influence on the intestinal levels of six species of bifidobacterial (*B. adolescentis, B. bifidum, B. breve, B. catenulatum, B. longum,* and *B. infantis*) and *Ruminococcus* spp. Total HMOs content ranges from about 15.0 g/L in group 1 to about 5.0 g/L, in group 4, because of the presence or absence of specific fucosyl-oligosaccharides. The study shows that unequal composition of group 1, 2, and 3 milks was not related to substantial differences in bifidobacterial species composition within infants fed group 1, 2, and 3 milks. However, in group 4 milk (with a slight quantity of fucosyloligosaccharides), the microbiota was characterized by a higher frequency of *Bifidobacteria adolescentis* and the absence of *Bifidobacteria catenulatum* and by harboring a different intestinal microbiota.

Early-life gut microbiome development is intrinsically linked to the risk of necrotizing enterocolitis (NEC) in preterm newborns. Masi’s study [16] integrates HMOs and infant intestinal metagenome data. Regardless of secretor status, the concentration of a single HMO, DSLNT, was lower in milk received by infants who exhibited abnormal microbiome development and developed NEC. A distinctive effect was observed when comparing different HMO groups: positive correlations were observed between sialylated HMO and *B. breve* and non-fucosylated/non-sialylated HMO and the *B. longum* group. In addition to the associations between HMO and bifidobacteria, positive correlations were observed between fucosylated HMO and *Akkermansia muciniphila* and between fucosylated/sialylated HMO and *Staphylococcus aureus*.

### 4.2. What Impact Does the HMO Profile Have on the Infant Gut Microbiota?

Several studies included in this review studied the relationship between HMO profiles and the gut microbiota [17,20,29,30]. Wang et al. [30] have analyzed the most abundant HMOs in the third month of lactation by HPLC-MS, and the fecal microbiota was performed by pyrosequencing of 16S Ribosomal RNA gene. The partial least squares regression of HMOs and microbiota showed that the HMO profile could predict infant fecal bacterial genera. *Bifidobacterium* was positively linked with the presence of LNFP I, MFLNH III, LSTb, and DSLNT and negatively linked with the presence of 2′FL and LDFT in human milk. Furthermore, most HMOs were associated with multiple bacterial genera; for example, 2′FL was positively linked with *Bacteroides* but negatively linked with *Bifidobacterium*, *Enterococcus*, *Veillonella,* and *Rothia*.

De Leoz’s work [29] only analyzes HMOs in fecal samples. Serial samples of two healthy infants were analyzed for several months, one of them only BF and the other fed first with formula supplementation for 4 days and then only BF. The samples were analyzed by bacterial DNA sequencing to characterize the microbiota and by mass spectrometry to determine the abundance of specific HMOs that passed through the intestinal tract without being consumed by the luminal bacteria. In both babies, the fecal bacterial population changed from microbes that do not consume HMO (*Enterobacteriaceae* and *Staphylococcaeae*, respectively) to bacteria that consume HMO during the first weeks of life (*Bacteroidaceae* and *Bifidobacteriaceae*), coinciding with the decrease in fecal HMO. These results support the concept that one function of HMOs is to selectively enrich a saccharolytic bacterial consortium despite the variety of bacteria introduced into the infant in the early days of life. As in the Wang et al. work [30], positive and negative correlations between the fecal isomers of HMO and the relative abundance of bacterial taxa were found at the order level. For example, MFLNH I and LnNH had opposite effects on the relative abundance of *lactobacillales* and *bifidobacteriales*.

Borewizc et al. [20] analyzed 121 pairs of mothers and their 1-month-old, healthy and breastfed children to investigate the association between selected maternal HMOs and the composition of the infant’s fecal microbiota. HMOs from the milk and feces of children were detected by mass spectrometry. Total HMO concentrations (between 2.0 and 6.5 mg mL-1) were lower than in other studies [21,24,31]. The Se+ mothers had many 2′FL and LNFP I, while they were absent in the Se-. Other structures, such as LNFP II, were present as major HMOs in the Le + mothers while they were absent in the Lewis negative mothers. They were unable to detect LNDFH I and DFL in breast milk samples containing α1-2- and α1-4-fucosis from mothers who were Le or Se negative. They detected some neutral HMOs in all milk samples, i.e., 3′FL, LNFP III, LNT, and LNnT. Only one mother lacked NHL and NHL in her milk. Acidic HMOs were detected in all milk samples. There was also a large variation in HMO concentrations in the infants’ stools. The fecal microbiota was characterized by amplicon sequencing of the Illumina HiSeq 16S rRNA gene, and infants were classified into three distinct microbial cluster types based on genus-level microbial abundance data using Dirichlet Multinomial Mixture (DMM) modeling I [60]. As in Wang’s work [30], which associated LNFPI positively with *Bacteroides* and *Bifidobacterium* and 2′FL with *Bacteroides*, Borewizc et al. [20] associated these two HMOs (very important in Se+ mothers) with the type A mixed microbiota group, which is characterized by a relative abundance of *Bacteroides* and *Bifidobacterium*. None of the other HMOs showed a significant association with the composition of the microbiota. Degradation of specific HMOs could be correlated with an increase in relative abundance of various phylotypes (OTUs) within the genus *Bifidobacterium* and to a lesser extent within the genera *Bacteroides* and Lactobacillus. A possible combined effect of various HMO structures may be necessary to guide the development of the microbiota in early life, or stronger associations may develop over a longer period, and at one month of age, the microbial profile of the infants in the study was still largely in its transitional phase.

The same authors developed a longitudinal study [17] investigating the association between these HMOs’ concentrations of breast milk and infant fecal microbial composition during the first three months of life. A smaller cohort of 24 pairs of Dutch mothers and healthy, breastfed, full-term babies was studied by analyzing the samples at 2, 6, and 12 weeks postpartum. A total of 18 very abundant HMOs (13 neutral and five acidic HMOs) were quantified by HPLC with MS and pulsed amperometric detection. The microbiota of the breastfed infants was studied by 16S rRNA region V4 sequencing. The HMOs decreased with the time of lactation, which corresponds to what was found in all the longitudinal studies of this review [17,22,24,25,28,29]. Only the concentrations of 3′FL and LNFP III were stable throughout lactation. The composition of the microbiota varied over the weeks and was associated with the mode of delivery and with the concentration of LNFP III at two weeks, with the sex of the infant, the mode of delivery, and the concentrations of 3′SL at six weeks, and with infant sex and LNH at 12 weeks of age. At any sampling point, strong relationships were found between individual breast milk HMO levels and OTUs relative abundance in infant feces, including the more predominant OTUs, *Bifidobacterium*. As in the work [29], the HMO concentrations in fecal samples decreased with age and were strongly and negatively correlated with the relative abundance of *Bifidobacterium*, but also *Parabacteroides*, *Escherichia*-*Shigella*, *Bacteroides*, *Actinomyces*, *Veillonella*, *Lachnospiraceae,* and *Erysipelotrichaceae*, indicating the probable importance of these taxa for HMO metabolism in vivo. Authors considered that HMO composition is only one of many factors regulating the colonization and structure of the infant GI microbial community.

### 4.3. Can Specific HMOs Be Added to Formula to Modulate the Infant Microbiota?

Two studies with oligosaccharide-supplemented infant formulas have also been included in the present review, one of them with a GOS (OM55N) [26] and another with two HMOs present in breast milk (2′FL and LNn) [18]. We have not included studies with bovine-milk-derived oligosaccharides [61].

Matsuki [26] investigated the effect of an infant formula supplemented with galactooligosaccharides (GOS; OM55N) on the growth of indigenous bifidobacteria. This was a randomized, double-blind, placebo-controlled trial with 35 babies. Fecal samples were taken at the start of the trial and 2 weeks later. They were analyzed by 16S metagenome analysis. After 2 weeks, although the GOS concentration was relatively low, the abundance of Bifidobacteriaceae was significantly higher and the Shannon index was significantly decreased in the GOS feeding group compared to the control.

Berger’s [18] study is a multicenter, controlled, double-blind, randomized clinical trial with healthy term infants who received an infant formula (control) or the same formula with 2′FL and LNn (trial) from enrollment (0–14 days) until 6 months. Then, all the babies received the same follow-up formula without HMO until they were 12 months old. BF infants served as the control group. The fecal microbiota was analyzed at 3 and 12 months by 16S rRNA gene sequencing. The microbiota of formula-fed 3-month-old infants was different if they received HMOs and closer to the microbiota of BF infants (for the microbial diversity, the global composition at the genus level, and the abundance of several major genera typical of that age period-increase of bifidobacterial-). The addition of 2′FL and LNnT to infant formula shifts the microbiota toward the microbiota observed with breastfeeding, the standard in infant nutrition. They also observed a decrease of *Escherichia* and unclassified *Peptostreptococcaceae*, a family to which *Clostridium difficile* belongs.

## 5. Conclusions

The oligosaccharides are quantitatively the third component in breast milk. However, the child cannot assimilate them as nutrients. They are prebiotics, but not all bacteria have the required enzymes to metabolize them. HMOs facilitate the establishment of a highly specialized microbial ecosystem, allowing the settlement of certain species and displacing others. In this review, we have verified that in most of the in vivo studies published since 2011, this prebiotic effect has been demonstrated for selected bacterial species such as certain *Bifidobacteria* and *Bacteroides*.

The longitudinal studies reviewed coincided in pointing out that the concentration of HMOs in breast milk varies during lactation, and this could determine the gradual development of the microbiota in the gastrointestinal tract. The MSeS largely determines the composition of HMOs and their concentrations in breast milk. Most of the investigations relate significant differences in the intestinal microbiota when comparing children breastfed by Se+ or Se- mothers. The MSeS can be, also, an important factor among babies with compromised microbiota, since it can match the intestinal microbiota of a child by vaginal delivery or cesarean section. Only two studies found no significant differences in the gut microbiota comparing children breastfed by Se+ or Se- mothers. Most studies state that babies fed Se+ generally had higher relative amounts of *Bifidobacterium*. The MSeS showed its influence on the composition and function of the infantile intestinal microbiota even at two years of age.

Regardless of MSeS, a lower concentration of a single HMO, DSLNT, could be the cause of premature infants exhibiting abnormal microbiome development and developing NEC. Beyond maternal genetics, there is individual variability in the presence of certain HMOs and their concentrations in breast milk. These differences also determine differences in the infant gut microbiota. Several investigations included in this review studied the relationship between HMO profiles and the gut microbiota. Positive and negative correlations were found between HMOs and the relative abundance of bacteria. However, in some cases, the relationships are contradictory, and some authors believe that the composition of HMO is only one of many factors that regulate colonization and the structure of the infant gastrointestinal microbial community. Likewise, the possibility of increasing bifidobacteria in risk populations, such as premature babies, is conceived by adding specific glycans. This was studied in two articles included in this review, and it was found that it displaces the microbiota towards the microbiota observed with breastfeeding, the standard in infant nutrition. Adding HMOs to infant formulas will bring them closer to breastfeeding outcomes. However, in addition, supplementation with certain HMOs could be considered in children fed by Se- mothers.

## Figures and Tables

**Figure 1 foods-10-01429-f001:**
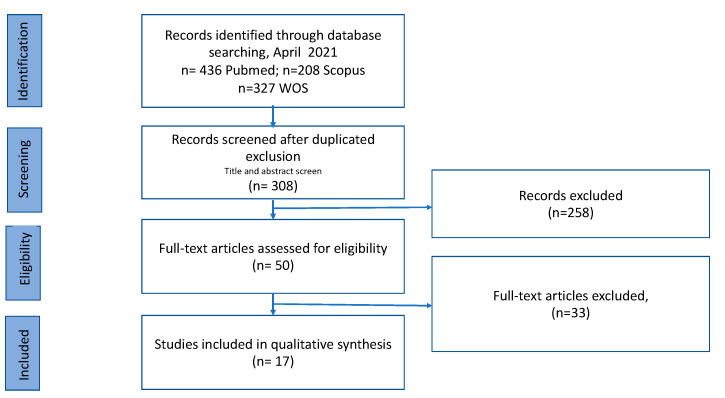
Flow diagram describing study selection process.

**Table 1 foods-10-01429-t001:** Characteristics and findings of included articles of included studies (n = 17).

Reference	Study Design	Population and Sample Number	HMO BM Analysis	Microbiota Analysis ^1^	Outcome Observed (Yes/No)	Main Finding
Tonon et al. [15]	Cross-sectional study	n = 48 pairs of mothers and C-sec and vaginally born BF infants. Sampling at one month postpartum.	HPLC-MS -MSeS: Occurrence of 4 α1-2 fucosylated HMO.	- QIAamp DNA Stool Mini Kit (Qiagen) - 16S rRNA region V4 (F515/R806) - 500-cycle Miseq V2 Kit and Miseq sequencing system (Illumina) - SILVA database	Yes	MSeS positively equates the intestinal flora of a child born vaginally or by C-sec.
Masi et al. [16]	Cohort study	n = 70 preterm infants (BF or FF-BF + LaBiNIC) (n = 33, NEC) Validation subset: (n = 48: n = 14, NEC).	HPLC-MS -19 most abundant HMOs. -MSeS: presence/absence 2′FL.	- DNeasy PowerSoil Kit (Qiagen) - Metagenomics - HiSeq X Ten sequencing system (Illumina) - MetaPhlan2 marker gene database	Yes	Independent of Se status, DSLNT concentration was lower in milk received by infants who showed abnormal microbiome development and developed NEC.
Borewicz et al. [17]	Cohort longitudinal Study	n = 24 mother–BF infant pairs. Sampling at 2, 6, and 12 weeks post-partum.	PGC-UPLC-MS and HPAEC-PAD -Total and relative abundance of 18 HMOs.	- Maxwell 16 Total RNA system (Promega) with Stool Transport and Recovery Buffer STAR (Roche) - 16S rRNA region V4 (F515/R806) - HiSeq 2000 sequencing system (Illumina) - SILVA database	No	Did not observe strong and consistent positive correlations between the HMOs and specific microbial OTUs, including *Bifidobacterium*. HMO composition is only one of many factors regulating infant gut microbial community.
Berger et al. [18]	Randomized double-blinded controlled multicentric clinical trial	n= 175 healthy term FF infants, (n = 87, IF with 2′FL and LNnT and n = 88, normal IF) Control: n = 38 BF infants. Sampling at 3 months post-partum.		- QIAamp DNA Stool minikit (Qiagen), plus FastPrep apparatus and Lysing Matrix B tubes (MP Biochemicals) - 16S rRNA regions V3 and V4 (S-D-Bact-0341-b-S-17/S-D-Bact-0785-a-A-21) - Miseq reagent kit V3 and Miseq sequencing system (Illumina) - Ribosome Database Project and Silva database	Yes	The addition of two very specific HMOs (2′FL and LNnT) to IF shifts the microbiota toward the microbiota observed with breastfeeding, the standard in infant nutrition.
Quin et al. [19]	Cohort study	n = 109 mother–BF infant pairs. Sampling at 5 months post-partum.	CE-LIF and HPLC-MS. -MSeS: Relative abundance 2′FL, LDFT, and LNFP1. -16 HMOs bearing sulfate and/or phosphate groups. -Nonsulfonated HMOs: 2′FL, LSTc, and LNP1	- QIAamp DNA Stool minikit (Qiagen). - 16S rRNA regions V3 and V4 (341F/805R) - MiSeq sequencing system (Illumina) - GreenGenes database.	Yes	Maternal genetics have a defining role in the establishment of early colonizers (abundance of *Enterobacteriaceae* was associated with MSeS), but maternal dietary intake during lactation appears to influence the community composition of the infant microbiome.
Borewicz et al. [20]	Cohort study	n = 121 healthy, full-term BF infants. Sampling: At, approximately, 1 month postpartum.	UPLC-MS -11 neutral and five acidic HMOs) PAEC-PAD -3′FL	- QIAamp DNA Stool minikit (Qiagen) - 16S rRNA region V4 (F515/R806) - HiSeq sequencing system (Illumina) - Silva database	Yes	Statistically significant associations between infant fecal microbiota composition and LNFPI and 2′FL levels. Degradation of specific HMOs could be correlated with an increase in relative abundance of various phylotypes within the genus *Bifidobacterium* and to a lesser extent within the genera *Bacteroides* and *Lactobacillus.*
Paganini et al. [21]	Double-masked randomized controlled trial study. Cross-sectional study.	n = 80 mother–infant pairs, BF + supplement or not Sampling at baseline of clinical trial, after 3 weeks and after 4 months.	HPAE-PAD -MSeS: presence/absence 2′FL and LNFPI - Total fucosylated sum of 2′FL, 3′FL, LNFPI, LNFPII, and LNFPIII; total sialylated sum of 6′SL, 3′SL, LSTd, LSTa, DSLNT and total non-fucosylated and non-sialylated sum of LNnT, LNT, and LNnH.	- Maxwell 16 Tissue LEV Total RNA Purification Kit (Promega) - 16S rRNA region V3–V4 (357F/802R) - MiSeq sequencing system (Illumina) - GreenGenes database	No	MSeS does not have a major impact on the gut microbiota of the mothers with the exception of a higher abundance of *C. perfringens* among Se- compared to Se mothers.
Bai et al. [22]	Longitud Cohort study	n = 56 mother–vaginally born BF pairs. Sampling at days 6, 42, 120, and 180 post-birth.	LC-QTOF-MS -MSeS: presence/absence LDFT and LNFP I	- E.Z.N.A. stool DNA kit (Omega Bio-tek) - 16S rRNA region V4 (520F/802R) - IlluminaGAIIx platform (Illumina) - NCBI NR database	Yes	*Bifidobacterial* established earlier (and in higher amounts in Se+-fed infants). The relative abundances of this genus continued to increase more than 180 days of lactation in the Se+ group.
Korpela et al. [23]	Cohort study	n = 76 mothers–C-sec and vaginally born BF infants. Sampling: BM on day 3 and feces at 3 months.	HPLC-MALDI-TOF and HPAEC -MSeS: 2′FL quantification.	- Repeated Bead Beating protocol and QIAamp DNA Stool Mini Kit columns (Qiagen) - 16S rRNA region V3–V4 (N.S.) - MiSeq sequencing system (Illumina) - SILVA database	Yes	The C-sec born infants of Se+ mothers had a more modest deviation in microbiota composition, compared to those of Se- mothers.
Davis et al. [24]	Longitudinal sub-study embedded within a randomized trial.	n = 33 mother–BF infant pairs. Sampling: At 4, 16, and 20 weeks postpartum.	HPLC-TOF. MSeS: α 1-2 fucosylated HMO quantification of 2′ FL, LDFT, TFLNH, DFLNHa, DFLNHc, and IFLNH I.	- Zymo ZR Fecal DNA MiniPrep™ Isolation Kit - 16S rRNA V4 region (F515/R806) - MiSeq 2000 sequencing system (Illumina) - Database not specified	Yes	The microbiome’s ability to break down certain types of oligosaccharides depends on the specific strains that make up the baby’s microbiota. These strains’ variability may contribute to their inability to find functional differences in microbiomes between babies fed by mothers of different secretor status.
Underwood et al. [25]	Cohort study	n = 29 preterm BF infants supplemented with *B. breve*, strain M16-V. Sampling: close to the probiotic start and 3 weeks later.	Nano-HPLC-chip/TOF-MS -MSeS: α(1,2) fucosylated HMOs abundance >6%.	- QIAGEN Stool Mini Kit - 16S rRNA region V4 (F515/R806) - MiSeq sequencing system (Illumina) - GreenGenes database	No	MSeS was not a significant predictor of response to the administered probiotic *B. breve.*
Matsuki et al. [26]	Randomized, double-blind, placebo-controlled trial	n = 35 FF infants (supplemented with GOS (OM55N). Sampling at the start of the trial and 2 weeks later.		- FastPrep FP 120 instrument and phenol/chloroform/isoamyl alcohol extraction - 16S rRNA V1–V2 regions (66F-TAG-linker A/338Rm-linker B) - 454 GS Junior platform (Roche) - Ribosomal Database Project	Yes	The formula supplementation with GOS (OM55N) stimulated the growth of *bifidobacteria* and resulted in reduced α-diversity of the gut microbiota.
Smith-Brown et al. [27]	Cohort study	n = 37 BF children 2 years old and 17 eligible mothers (20 excluded due to pregnancy within the previous 12 months)	MSeS was determined from blood and saliva samples using hemagglutination inhibition technique.	- Bead beating and Maxwell 16 Tissue DNA Purification Kit (Promega) - 16S rRNA region V6–V8 (1406F/1525R) - MiSeq sequencing system (Illumina) - GreenGenes database	Yes	*Bifidobacterium* was increased in the BF children of Se+ mothers compared to Se- mothers.
Lewis et al. [28]	Longitudinal cohort study	n = 44 mother–BF infant pairs Sampling: At day 6, 21, 71, and/or 120 postpartum.	Nano-HPLC-chip-TOF-MS. -MSeS: α(1,2) fucosylated HMOs abundance.	- ZR Fecal DNA MiniPrep kit (ZYMO) - 16S rRNA region V4 (F515/R806) - MiSeq sequencing system (Illumina) - Ribosomal Database Project	Yes	Se+ fed infants generally had higher relative amounts of *Bifidobacterium* and *Bacteroides* and lower levels of enterobacteria, clostridia, and streptococci.
De Leoz et al. [29]	Longitudinal Proof-of-concept study	n = 2 infants (A: BF; B: formula supplementation 4 days and then was solely BF) Sampling: twice/week first month, twice/month second month, and once or twice/month thereafter.	Nano-HPLC-Chip/TOF MS -HMO profile and HMO quantitation to the isomer level (fecal samples).	- QIAmp DNA Stool Mini Kit (Qiagen) - 16S rRNA regions V1–V3 and V4 (B-8F/A-518R and F515/R806) - Genome Sequencer GS-FLX (Roche) and Genome Analyzer II sequencing system (Illumina) - Ribosomal Database Project	Yes	Fecal HMO profiles correlated with changes in bacterial population. Positive and negative correlations between the fecal isomers of HMO and the relative abundance of bacterial taxa were found at the order level.
Wang et al. [30]	Quasi-experimental cohort study	n = 22 mother–infant pairs. (16 BF and 6 FF) Sampling: At 3 months post-partum.	HPLC-Chip/TOF-MS -Until 141 HMOs/sample.	- QIAgen DNA Mini Stool Kit (Qiagen) - 16S rRNA region V1-V3 (27F-DegS/534R) - 454 Life Sciences Genome Sequencer FLX (Roche) - Ribosomal Database Project	Yes	The microbial composition of BF infants is correlated with the presence of HMO in their mother′s milk.
Coppa et al. [31]	Cohort study	n = 256 mother–infant pairs. Sampling: At 30 days post-partum.	HPAEC -18 HMOs. -4 BM groups on the basis of the presence or the absence of 2′FL and LNFPII.	- Phenol/chloroform extraction - PCR-DGGE (*B. adolescentis, B. catenulatum*, *B. infantis*, *B. longum, B. breve*, and *B. bifidum* primers) - PCR fragment sequencing - Ribosomal Database Project	Yes	No substantial differences in *bifidobacteria* species composition within infants fed with groups 1, 2, and 3 BM; with group 4 BM (with slight quantity of fucosyloligosaccharides), the microbiota was characterized by a greater frequency of *B. adolescentis* and the absence of *B. catenulatum* and harbored a different intestinal microbiota.

Breastfed (BF); Infant Formula (IF); Formula-fed (FF); Cesarean section (C-sec); Necrotizing enterocolitis (NEC); Operational taxonomic units (OTUs); Breastmilk (BM); Maternal secretor status (MSeS); Porous graphitized carbon-ultra high-performance liquid chromatography–mass spectrometry (PGC-UPLC-MS); High-performance anion exchange chromatography with pulsed amperometric detection (HPAEC-PAD); Capillary electrophoresis with laser-induced fluorescence (CE-LIF); High-performance anion exchange chromatography–pulsed amperometric detection (HPAEC-PAD); Matrix-assisted laser desorption/ionization–time of fight mass spectrometry (MALDI-TOF-MS); *Bifidobacterium*-specific terminal restriction fragment length polymorphism assay (Bif-TRFLP); LaBiNIC (*L. acidophilus*, *B. infantis,* and *B. bifidum*) or Infloran (*L. acidophilus* and *B. Bifidum*).^1^ Including the following information: DNA extraction kit, 16S rRNA gene hypervariable region (primers), sequencing system, rRNA database. N.S.: not specified.

## Data Availability

Not applicable.

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
