# Peer review of "Human Milk Oligosaccharides (HMOs) and Infant Microbiota: A Scoping Review"

_foods, 2021, doi:10.3390/foods10061429_

Round 1

Reviewer 1 Report

In this manuscript, 17 articles were selected as the main research objects, and the influence of mother's different genotypes on HMO profiles, the influence of some factors on infant intestinal microbiota were discussed. The comments are below:

  1. There are many reviews about the effect of HMOs on infant intestinal microbiota, but the innovation and highlights of this review are not obvious. Here are some references, but not limited to the following:[2]Pamela, Thomson, Daniel, et al. Human milk oligosaccharides and infant gut bifidobacteria: Molecular strategies for their utilization.[J]. Food Microbiology, 2017.[4]Hegar, B Wibowo Y, Basrowi, RW. The role of two human milk oligosaccharides, 2′-Fucosyllactose and Lacto-N-Neotetraose, in infant nutrition[J]. Pediatric Gastroenterology Hepatology & Nutrition, 2019, 22(4): 330.
  2. [3]Jost T, Lacroix C, Braegger C, Chassard C. Impact of human milk bacteria and oligosaccharides on neonatal gut microbiota establishment and gut health, Nutrition Reviews, 2015, 73(7): 426-437.
  3. [1]Walsh C, Lane JA, Sinderen DV, et al. Human milk oligosaccharides: Shaping the infant gut microbiota and supporting health[J]. Journal of Functional Foods, 2020, 72:104074.
  4. In the “4. Discussion” part, the logic between some paragraphs is not strong, and the core of each paragraph could not be accurately understood. My suggestion is to write a sentence at the beginning of each paragraph to play a connecting role. For example, line 402.
  5. Line 335: “B. infantis was the only (subs) species positively correlated with LNnT, confirming previous reports particular HMO species enhance B. infantis growth.” Is B. infantis the only bacteria positively related to LNnT? Could LNnT inhibit all bacteria except B. infantis? This is inaccurate description. Please provide more literature to support your statements.
  6. The manuscript is divided into “Materials and methods, Results and Discussion”. It is meaningless.
  7. Line 20: There is a reference in the abstract, but generally there is no such situation.
  8. Line 15: Due to this and other characteristics HMOs are considered prebiotic agents. What are other characteristics?
  9. Line 20: Differences in HMOs profiles have been related to breast milk and the infant's gut microbiota. This is an unclear description, please correct.
  10. Line 23: infant microbiome. The range of infant microbiome is large, manuscprit only describes the infant intestinal microbiota.
  11. Line 25: human milk, breast milk. What's the difference between them?
  12. Line 65: other types of butyrate and propionate producing microbiota. The microbiota needs to be explained in detail.
  13. Line 67-69: pHMOs did not affect the growth of Escherichia coli (UPEC) Pseudomonas aeruginosa, or Staphylococcus aureus.
  14. As described in reference [9], pHMOs only inhibited the proliferation of Streptococcus agalactiae, Escherichia coli (UPEC), and Pseudomonas aeruginosa, and Staphylococcus aureus were not inhibited. The expression is not rigorous, and the authors should list the inhibition of HMO on a variety of pathogens in detail.
  15. Line 175: Please check all the references in the manuscript. In reference [22], there was not a method of high performance liquid chromatography (HPLC) with ultraviolet (UV) to analyze the profile of HMOs.
  16. Line 187: 16 HMOs have been quantified, it would be better for the authors to add a table to clearly show the content of various HMOs.
  17. Line 200: One study analyzed the HMOs only in fecal samples. There was no references.
  18. Line 386-390: Alongside Neu5Gc, the study shows that fucose levels in breast milk are associated with Bacteroides and Escherichia spp. in infant stool. Finally, and unexpected given that most Lactobacillus spp. do not grow well on HMOs, research found that Lactobacillus spp. in infant stool is correlated with total galactose concentration in sulfonated milk oligosaccharides. There was no references.
  19. Line 201-205: What is the purpose of this paragraph? It seems that it is not directly related to the conclusion of this paper.
  20. Line 255: the individual HMOs, except DSLNT, are different in the Se+ state. Does it refer to the content?
  21. Line 261: Please check the brackets.
  22. Redundant spaces appeared in the article, such as Line 260, 261, 277.
  23. Line 239: there was no space in influencingHMOs.
  24. Line 418: Bifidobacterium was negatively linked with the presence of 2’-FL and LDFT in human milk. There are many literatures about the promoting effect of 2'- FL on Bifidobacterium. How to explain?

Author Response

Resposes to Reviewer 1:

In this manuscript, 17 articles were selected as the main research objects, and the influence of mother's different genotypes on HMO profiles, the influence of some factors on infant intestinal microbiota were discussed. The comments are below:

  1. There are many reviews about the effect of HMOs on infant intestinal microbiota, but the innovation and highlights of this review are not obvious. Here are some references, but not limited to the following:[2]Pamela, Thomson, Daniel, et al. Human milk oligosaccharides and infant gut bifidobacteria: Molecular strategies for their utilization.[J]. Food Microbiology, 2017.[4]Hegar, B Wibowo Y, Basrowi, RW. The role of two human milk oligosaccharides, 2′-Fucosyllactose and Lacto-N-Neotetraose, in infant nutrition[J]. Pediatric Gastroenterology Hepatology & Nutrition, 2019, 22(4): 330.

Response: The reviewer indicates that there are many reviews on the effect of HMOs on the infant gut microbiota, however, most of them review some effects of singular HMOs (2′-Fucosyllactose and Lacto-N- Neotetraose) in limited aspects of the gut microbiota (bifidobacteria, in most cases) and includes other factors such as bacteria in breast milk.

  1. In the “4. Discussion” part, the logic between some paragraphs is not strong, and the core of each paragraph could not be accurately understood. My suggestion is to write a sentence at the beginning of each paragraph to play a connecting role. For example, line 402.

Response: The discussion has been redone to organize the text in a more understandable way.

  1. Line 335: “B. infantis was the only (subs) species positively correlated with LNnT, confirming previous reports particular HMO species enhance B. infantis growth.” Is B. infantis the only bacteria positively related to LNnT? Could LNnT inhibit all bacteria except B. infantis? This is inaccurate description. Please provide more literature to support your statements.

In Davis's work, B. infantis was the only positively related bacteria (in a statistically significant way) with LNnT. B. infantis encodes several enzymes to metabolize the major glycans in breast milk. This fact indicates that some bifidobacteria are better equipped for HMO consumption than others. There is a selective correspondence between what mothers secrete in their milk and what this strain consumes. But, in no case LNnT inhibits all bacteria except B. infantis.

As Davis indicates, B. infantis possesses a large genetic complement of enzymatic tools to consume the major human milk glycans in breast milk, suggesting that the dominance of the gut community by this microbe is driven by the availability of these substrates.

More bibliography has been included.

  1. The manuscript is divided into “Materials and methods, Results and Discussion”. It is meaningless.

The manuscript is divided into Methodologies, Results, Discussion and Conclusions. 

  1. Line 20: There is a reference in the abstract, but generally there is no such situation.

The reference in the abstract has been removed. 

  1. Line 15: Due to this and other characteristics HMOs are considered prebiotic agents. What are other characteristics?

For a better understanding, the abstract has been modified…Human milk oligosaccharides (HMOs) are the third most abundant solid component of breast milk. However, the newborn cannot assimilate them as nutrients. They are recognized prebiotic agents (the first in the newborn diet) that stimulate the growth of beneficial microorganisms, mainly the genus Bifidobacterium, dominant in the gut of breastfed infants. 

  1. Line 20: Differences in HMOs profiles have been related to breast milk and the infant's gut microbiota. This is an unclear description, please correct.

For a better understanding, the abstract has been modified… Differences in the profiles of HMO have been linked to breast milk microbiota and gut microbial colonization of babies.

  1. Line 23: infant microbiome. The range of infant microbiome is large, manuscprit only describes the infant intestinal microbiota.

For a better understanding, the abstract has been modified…Here we provide a review of the scope of reports on associations between HMOs and the infant gut microbiota, to assess the impact of HMO composition. 

  1. Line 25: human milk, breast milk. What's the difference between them?

They are synonymous terms. Human milk has been removed from key words. 

  1. Line 65: other types of butyrate and propionate producing microbiota. The microbiota needs to be explained in detail.

For a better understanding, the text has been modified……The partial metabolization of oligosaccharides gives rise to "postbiotic" compounds that stimulate the growth of other types of butyrate and propionate producing microbiota. These short-chain fatty acids have a trophic effect on the intestinal barrier, stimulating mucin release and modulating the immune system, promoting immune tolerance [11]…. 

  1. Line 67-69: pHMOs did not affect the growth of Escherichia coli (UPEC) Pseudomonas aeruginosa, or Staphylococcus aureus. As described in reference [9], pHMOs only inhibited the proliferation of Streptococcus agalactiae, Escherichia coli (UPEC), and Pseudomonas aeruginosa, and Staphylococcus aureus were not inhibited. The expression is not rigorous, and the authors should list the inhibition of HMO on a variety of pathogens in detail.

The text has been revised and we hope it is more understandable...The HMOs-consuming bacteria also inhibit that pathogenic bacterium colonize the intestine by reducing nutrient availability and the production of antimicrobial substances. A direct bacteriostatic action of oligosaccharides in breast milk has been demonstrated in the case of group B Streptococcus, which cannot proliferate in a medium with specific non-sialylated HMOs [12]… 

  1. Line 175: Please check all the references in the manuscript. In reference [22], there was not a method of high performance liquid chromatography (HPLC) with ultraviolet (UV) to analyze the profile of HMOs.

This is a mistake. It is a randomized controlled clinical trial with galacto-oligosaccharides (GOS). Human milk is not tested for HMO content. 

  1. Line 187: 16 HMOs have been quantified, it would be better for the authors to add a table to clearly show the content of various HMOs.

For a better understanding, the text has been modified ….Apart from MSeS, eight studies quantified individual HMOs [16,17,19–21,29–31].  Quin et al. [19], also analyzed HMOs containing sulfate and / or phosphate groups… 

  1. Line 200: One study analyzed the HMOs only in fecal samples. There was no references.

The reference was introduced….One study analyzed the HMOs only in fecal samples [29].

  1. Line 386-390: Alongside Neu5Gc, the study shows that fucose levels in breast milk are associated with Bacteroides and Escherichia spp. in infant stool. Finally, and unexpected given that most Lactobacillus spp. do not grow well on HMOs, research found that Lactobacillus spp. in infant stool is correlated with total galactose concentration in sulfonated milk oligosaccharides. There was no references.

The reference was introduced….….infant stool is correlated with total galactose concentration in sulfonated milk oligosaccharides [19]. 

  1. Line 201-205: What is the purpose of this paragraph? It seems that it is not directly related to the conclusion of this paper.

According to reviewer 3, we have expanded the paragraph in the new version of the manuscript….Sequencing directed at different validated hypervariable regions of the 16S rRNA gene was used in most of the studies: not reported [25], V1-V2 [26], V3 [18], V4 [15,17,20,22,24,28], V3-V4 ([19,21,23] or V6-V8 [27]. In Masi's study [16], metagenomic sequencing was performed and in other research work the quantitative polymerase chain reaction (PCR) served as a technique for the analysis of bifidobacterial species [31]. The sequencing of 16S rRNA gene amplicons has been a very popular approach to assess microbial communities in feces and other human matrices in the last decades [32,33]. Human feces are high microbial load samples, and both sample processing and primer selection largely impact 16S gene-based profiling results [34–37]. In this context, the protocol to extract DNA should be selected upon not only biomass abundance but also the expected gut microbiota composition. For instance, DNA extraction with no bead-beating step have resulted in absence of bifidobacteria in the sequence data, even when using optimised primers [34]. In Table 1 an overview of different hypervariable regions of 16S rRNA gene that have been targeted in gut microbiota studies, using different primer pairs. The choice of 16S rRNA region can significantly affect the estimates of taxonomic diversity [38,39]. For instance, V2-V3 or V3-V4 regions compute similar number or reads per phyla but at lower taxonomic ranks the differences become larger [38]. Interestingly, literature indicates that bifidobacteria are often neglected by several common primer pairs [36]. For example, common primers targeting V1 region have usually poor coverage of Bifidobacterium, while those targeting V4 will likely cover Bifidobacterium but not Cutibacterium [33]. In order to overcome these discrepancies and avoid biases, a careful selection of DNA extraction protocol and primer pair is highly recommended… 

  1. Line 255: the individual HMOs, except DSLNT, are different in the Se+ Does it refer to the content?

For a better understanding, the text has been modified ….The absence of 2'FL and other fucosyl-HMOs explains the lower total amount of HMO in the milk of women without Se [5]. But also, all individual HMOs differed by Secretor status, except for disialyllacto-N-tetraose (DSLNT) [7]. 

  1. Line 261: Please check the brackets.

Redundant spaces appeared in the article, such as Line 260, 261, 277. 

The text has been revised. 

  1. Line 239: there was no space in influencing HMOs.

 The text has been revised.

  1. Line 418: Bifidobacterium was negatively linked with the presence of 2’-FL and LDFT in human milk. There are many literatures about the promoting effect of 2'- FL on Bifidobacterium. How to explain?

 Although there is much literature on the promoting effect of 2'FL on bifidobacteria, in Wang et al. study, the relative abundance of fecal Bifidobacterium was negatively linked to the presence of 2′FL and LDFT in human milk. Furthermore, Underwood study [25], found that  a strain of bifidobacterium (B. breve M16-V) is a selective consumer of human milk oligosaccharides consuming 3’FL and LNT but not 2'FL.

Reviewer 2 Report

Comments to the Author

MANUSCRIPT DETAILS

Ms. Ref. No.: foods-1242364

Title: Human milk oligosaccharides (HMOs) and infant microbiota: a scoping review

Article Type: Review Article

JOURNAL: Foods

GENERAL COMMENTS

This review aimed to provide reports on associations between HMOs and the infant gut microbiota, to assess the impact of HMO composition on the infant microbiome.

The interest in this manuscript is significant enough to merit publication.

My recommendation on submitted manuscript to Foods is to be accepted after minor revisions.

The comments and questions provided below may help the authors to put the manuscript into better appropriate form for publication.

SPECIFIC COMMENTS

- L20: It is not preferred to cite references in the Abstract. So kindly you can cite this reference in another suitable location in the MS.

- L78: I suggest naming Section 2 “Methodologies” as “Materials” are not represented in the common concept.

- L84: It is preferred to mention authors’ contributions under the title specified for this purpose, while methodology is preferred to be written in passive voice. Kindly revise thoroughly for similar comments.

- Table 1: Unify No not Non (French) nor Not.

- Grammar need to be revised thoroughly (e.g. L302 More studies find..), found.

- L519: Revise repeated words.

- Revise thoroughly to put scientific names in italics (e.g. L527).

- Add a status of “Authors’ contributions” at the end of the MS.

Author Response

The authors are grateful to the referee’s comments. In the revised version of the manuscript the aspects mentioned by the reviewer have been corrected.

Reviewer 3 Report

Please see attached comments.

Author Response

(The authors gave the same response as above.)

Round 2

Reviewer 1 Report

The manuscript has been improved as compared to its previous version but still does not expose enough novelty and requires several corrections. My additional comments are below.

  1. The authors put forward a viewpoint that DNA extraction protocol and primer pair may have a great influence on the composition of intestinal flora using 16S rRNA sequencing method. It is hoped that the authors could add the DNA extraction protocol and primer pair involved in each study in Table 1 for the reference of researchers who will do related experiments.
  2. Line 292-294:Furthermore, caesarean-born infants of Se+ mothers had significantly increased relative abundance of Verrucomicrobia (Akkermansia muciniphila), that can degrade HMOs [50]. What is the reference of this sentence? [23]? The reference only stated “Akkermansia was increased in the section-born infants of secretor mothers, supporting the suggestion that this organism may degrade HMOs.” But there was no description in the reference of the effect of the mode of delivery. And there are no relevant statements in reference [50] and [28].
  3. The authors mentioned “breast milk microbiota” more than once. Do the microbiota in breast milk come from around the breast or from breast milk itself? Maybe it's the former. If it's the former, the description of “breast milk microbiota” is inaccurate.
  4. Line 308-320: Only two studies showed that there was no significant difference in intestinal flora between Se+ and Se- breastfeeding infants. What is the purpose of supplementing Underwood's research [25]? There only stated that this study supplemented B. brevis for BF infants, did not show the results, and the study was not directly related to the above review.
  5. Line 279:4.1.2. "Considering the mode of birth" should be revised to 4.1.1.
  6. Some grammar problems:Line 498: “the fecal concentration of HMO” should be revised to “the HMOs concentrations in fecal”.
  7. Line 484: “breast milk concentrations of these HMOs” should be revised to “these HMOs concentrations of breast milk”.

Author Response

The authors are grateful to the referee’s comments. In the revised version of the manuscript the aspects mentioned by the reviewer have been corrected.

Responses to Reviewer 1:

The manuscript has been improved as compared to its previous version but still does not expose enough novelty and requires several corrections. My additional comments are below.

  1. The authors put forward a viewpoint that DNA extraction protocol and primer pair may have a great influence on the composition of intestinal flora using 16S rRNA sequencing method. It is hoped that the authors could add the DNA extraction protocol and primer pair involved in each study in Table 1 for the reference of researchers who will do related experiments.

Table 1 has been revised following reviewer’s suggestion.

  1. Line 292-294:Furthermore, caesarean-born infants of Se+ mothers had significantly increased relative abundance of Verrucomicrobia (Akkermansia muciniphila), that can degrade HMOs [50]. What is the reference of this sentence? [23]? The reference only stated “Akkermansia was increased in the section-born infants of secretor mothers, supporting the suggestion that this organism may degrade HMOs.” But there was no description in the reference of the effect of the mode of delivery. And there are no relevant statements in reference [50] and [28].

The text has been revised and we hope it is more understandable...Furthermore, caesarean-born infants of Se+ mothers had significantly increased relative abundance of Verrucomicrobia (Akkermansia muciniphila) [23], that can degrade HMOs [50]. This can strengthens the gut barrier and likely contributes positively to infant gut health [51]. …

  1. The authors mentioned “breast milk microbiota” more than once. Do the microbiota in breast milk come from around the breast or from breast milk itself? Maybe it's the former. If it's the former, the description of “breast milk microbiota” is inaccurate.

The microbiota in breast milk comes from around the breast and from breast milk itself. The term "breast milk microbiota" is accurate and widely used in the literature. In breast milk there is a microbiota of maternal origin and another that we could consider exogenous. Regarding microorganisms of maternal origin, bacteria from the gastrointestinal and oropharyngeal tracts could translocate and migrate to the mammary glands through an endogenous cellular pathway (the enteric and oropharyngeal tracts). The mammary gland is another source of microorganisms in which mastitis, previous pregnancies or even cancer intervene. Regarding the exogenous origin, the existence of a retrograde translocation of bacteria in the child has been postulated. This exogenous source must add to the contaminating flora of the utensils in contact with breast milk.

  1. Line 308-320: Only two studies showed that there was no significant difference in intestinal flora between Se+ and Se- breastfeeding infants. What is the purpose of supplementing Underwood's research [25]? There only stated that this study supplemented B. brevis for BF infants, did not show the results, and the study was not directly related to the above review.

Perhaps the authors have failed to sufficiently explain the relevance of the work of Underwood in this review. One aspect of the work reviewed was supplementation with B. brevis, but not the only one of relevance. This is an investigation with premature infants in which bifidobacteria are conspicuously absent even when the diet is exclusively human milk. Different species (and even subspecies and strains) of bifidobacteria vary in their ability to colonize the gastrointestinal tract, based largely on its ability to use HMOs as a source of nutrients.

The text has been revised ….Bifidobacteria are conspicuously absent even when the diet is exclusively human milk. In Underwood study [25], 29 preterm BF infants were supplemented with B. breve. Stool sampling was performed near the time of probiotic initiation and again 3 weeks later. An increase in Enterobacteriaceae over time was pronounced in this cohort. Children were divided into ‘‘responders’’ and ‘‘nonresponders’’ when having few bifidobacteria in the second stool sample. Nonresponders had significantly higher percentages of Enterobacteriaceae and Clostridiaceae than responders. Infants with secretor mothers, delivery type, and antibiotic treatment did not differ between responder and nonresponder infants. Higher percentages of total fucosylated HMOs and lower percentages of undecorated HMOs (those lacking both fucose and sialic acid) were found in the milk fed nonresponder babies. MSeS was not a significant predictor of response to the administered probiotic. Although B. breve M16-V is a selective consumer of human milk oligosaccharides (most strains consumes 3’FL and LNT but not 2'FL), the undecorated HMOs, aggressively consumed by all B. breve strains, it is what determines the differences.

  1. Line 279:4.1.2. "Considering the mode of birth" should be revised to 4.1.1.

OK

  1. Some grammar problems:Line 498: “the fecal concentration of HMO” should be revised to “the HMOs concentrations in fecal”.

OK

  1. Line 484: “breast milk concentrations of these HMOs” should be revised to “these HMOs concentrations of breast milk”.

OK
